# Gallium-Doped Hydroxyapatite Shows Antibacterial Activity against *Pseudomonas aeruginosa* without Affecting Cell Metabolic Activity

**DOI:** 10.3390/jfb14020051

**Published:** 2023-01-17

**Authors:** Marika Mosina, Claudia Siverino, Liga Stipniece, Artemijs Sceglovs, Renats Vasiljevs, T. Fintan Moriarty, Janis Locs

**Affiliations:** 1Rudolfs Cimdins Riga Biomaterials Innovation and Development Centre, Institute of General Chemical Engineering, Faculty of Materials Science and Applied Chemistry, Riga Technical University, Pulka 3, LV-1007 Riga, Latvia; 2Baltic Biomaterials Centre of Excellence, Headquarters at Riga Technical University, LV-1048 Riga, Latvia; 3AO Research Institute Davos, 7270 Davos, Switzerland

**Keywords:** calcium phosphate, antibacterial properties, gallium, biocompatibility

## Abstract

Calcium phosphates (CaPs) have been used in bone regeneration for decades. Among the described CaPs, synthetic hydroxyapatite (HAp) has a chemical composition similar to that of natural bone. Gallium-containing compounds have been studied since the 1970s for the treatment of autoimmune diseases and have shown beneficial properties, such as antibacterial activity and inhibition of osteoclast activity. In this study, we synthesized hydroxyapatite (HAp) powder with Ga doping ratios up to 6.9 ± 0.5 wt% using the wet chemical precipitation method. The obtained products were characterized using XRD, BET, FTIR, and ICP-MS. Ga^3+^ ion release was determined in the cell culture media for up to 30 days. Antibacterial activity was assessed against five bacterial species: *Pseudomonas aeruginosa*, *Escherichia coli*, *Staphylococcus aureus*, *Staphylococcus epidermidis*, and *Streptococcus pyogenes*. The biocompatibility of the GaHAp samples was determined in human fibroblasts (hTERT-BJ1) through direct and indirect tests. The structure of the synthesized products was characteristic of HAp, as revealed with XRD and FTIR, although the addition of Ga caused a decrease in the crystallite size. Ga^3+^ was released from GaHAp paste in a steady manner, with approximately 40% being released within 21 days. GaHAp with the highest gallium contents, 5.5 ± 0.1 wt% and 6.9 ± 0.5 wt%, inhibited the growth of all five bacterial species, with the greatest activity being against *Pseudomonas aeruginosa*. Biocompatibility assays showed maintained cell viability (~80%) after seven days of indirect exposure to GaHAp. However, when GaHAp with Ga content above 3.3 ± 0.4 wt% was directly applied on the cells, a decrease in metabolic activity was observed on the seventh day. Overall, these results show that GaHAp with Ga content below 3.3 ± 0.4 wt% has attractive antimicrobial properties, without affecting the cell metabolic activity, creating a material that could be used for bone regeneration and prevention of infection.

## 1. Introduction

Calcium phosphates (CaPs) are widely used in bone regeneration due to their unique properties, such as biocompatibility, osteoinductivity, and osteoconductivity [1,2,3]. They are often applied as bone cements [4], scaffolds [5], and coatings [1,2]. The biological activity of any CaP depends on the physicochemical properties, such as Ca/P molar ratio and solubility. Even though there is a great number of CaPs, hydroxyapatite (HAp) is the most extensively used one due to its similarity to the mineral component of bone. HAp is the most thermodynamically stable phase of CaP, with a characteristic Ca/P molar ratio of 1.67 and relatively low solubility. Numerous studies focusing on HAp have been undertaken in the last decades with both in vitro and in vivo assays, and conclusions have underlined its osteoconductive and osteoinductive properties [6,7,8,9,10,11,12,13].

In orthopedics, bacterial infection is a major problem leading to revision surgeries, implant removal, and possibly even amputation. Bacterial contamination of the wound may arise from the patient’s skin or from the surrounding environment [14,15,16,17]. Antibiotics have proven to be the gold standard in preventing and treating infections. However, the systemic administration of antibiotics can have a negative effect on the body (e.g., renal toxicity or microbiome dysbiosis), and it can lead to the potentially low active local concentration of antibiotic at the infection site. Additionally, antibiotic resistance of clinically relevant pathogens is increasing every year, which only emphasizes the need for the development of new biomaterials with improved antibacterial properties [18].

The doping of CaPs, including HAp, with different metal ions has often been performed to enhance the material properties, in terms of bioactivity and/or antibacterial properties [19]. HAp has the ability to incorporate isomorphic substituents, such as magnesium (Mg), sodium (Na), strontium (Sr), silver (Ag), copper (Cu), etc., due to its flexible and stable crystalline structure [20]. The addition of antibacterial ions such as Ag [21,22], Cu [23,24], Zn [25], and Ga [26] provides antibacterial assets to CaPs. Tailoring a biomaterial with the aforementioned capacities could help to prevent bacterial growth at the surgical site after implantation.

Gallium has been known since the 1970s, and next to its antibacterial potential, it has been used to treat bone diseases. Gallium nitrate (Ga(NO_3_)_3_) is used as a drug to treat hypercalcemia [27,28,29]. The antibacterial activity of gallium compounds, such as gallium maltolate, nitrate, and citrate has been reported against different bacterial species, such as *Mycobacteriaceae (M.) tuberculosis*, *Mycobacterium (M.) avium*, *Staphylococcus (S.) aureus*, *Escherichia (E.) coli* [30], and *Pseudomonas (P.) aeruginosa* [31,32]. To date, the antibacterial activity of gallium-doped hydroxyapatite (GaHAp) has primarily been assessed against *P. aeruginosa* [29,33]. The antibacterial activity of gallium against both Gram-positive and Gram-negative bacterial species is supported by the similarity between Ga^3+^ and Fe^3+^ ions [34]. Namely, Ga^3+^ replaces Fe^3+^ in bacteria, resulting in the disruption of protein metabolism and leading to bacterial death [16,32].

In this study the antibacterial activity of GaHAp synthesized using the wet precipitation method was tested for the first time against a wide range of bacterial species: *P. aeruginosa*, *E. coli*, *S. aureus*, *S. epidermidis*, and *S. pyogenes*. Additionally, the biocompatibility of the produced GaHAp was assessed on human fibroblasts (hTERT-BJ1).

## 2. Materials and Methods

### 2.1. Synthesis of the Gallium-Doped Hydroxyapatite (GaHAp)

Gallium-doped hydroxyapatite (GaHAp) was synthesized via wet chemical precipitation using calcium oxide (CaO; >98%; Jost chemical, St. Louis, MI, USA), orthophosphoric acid (H_3_PO_4_; 75%; “Latvijas ķīmija” Ltd., Riga, Latvia), and gallium nitrate hydrate (Ga(NO_3_)_3_·xH_2_O; 99.9% trace metal; Sigma-Aldrich, Burlington, MA, USA). In order to calculate the mass of gallium nitrate hydrate, the amount of water molecules of Ga(NO_3_)_3_·4.2H_2_O was determined with inductively coupled plasma mass spectrometry (ICP-MS; Agilent 7700X; Santa Clara, CA, USA). The gallium concentrations used in the synthesis were 2, 4, 6.3, and 8 wt% with respect to the theoretical HAp yield. The initial (Ca + Ga)/P molar ratio of the reagents was 1.67. The molar ratio was kept constant for all the concentrations. Three replicates of the synthesis were performed for each Ga concentration.

The synthesis processes were performed in the synthesis workstation EasyMax 102 Advanced (Mettler Toledo, Columbus, OH, USA). CaO powder was added to deionized water, under vigorous stirring (420 rpm) at room temperature (22 °C), in order to obtain Ca(OH)_2_ suspension. Then, Ga(NO_3_)_3_·4.2H_2_O powder was added and stirred for five minutes. The synthesis mixture was heated to 45 °C, and the temperature was maintained constant during synthesis. Then, 2M H_3_PO_4_ was added to the starting suspension of Ca(OH)_2_ and Ga(NO_3_)_3_·4.2H_2_O at an addition rate of 0.6 mL/min. The addition rate was reduced to 0.1 mL/min while approaching the synthesis end pH, 6.90 ± 0.05. The obtained precipitates were aged in the mother liquors at ambient temperature overnight (approximately 20 h). After ageing, the precipitates were vacuum-filtered and washed with 1 L of deionized water. HAp synthesized without Ga(NO_3_)_3_·4.2H_2_O (also known as pure HAp) was used as a reference.

The synthesized products were used in two forms depending on the performed test: as paste (filtered wet precipitates) or as dried powder. GaHAp paste was steam-sterilized in a table-top autoclave at 121 °C for 20 min. To obtain powder, the paste (sterilized or non-sterilized) was dried in an oven at 105 °C for 24 h. Dried agglomerates were crushed with a mortar and pestle to obtain fine powder. GaHAp paste was used in the indirect cytotoxicity tests, while GaHAp powder was used in physicochemical characterization, antibacterial assays, and direct cytotoxicity tests.

### 2.2. Characterization Methods

#### 2.2.1. X-ray Diffraction

The phase composition of the powders was analyzed using X-ray diffractometry (XRD; PANalytical X’Pert PRO; Westborough, MA, USA). XRD patterns were recorded using Ni filter and Cu Kα radiation at 40 kV and 30 mA, with a 2θ range of 10–70°.

The crystallite size was calculated from the X-ray diffraction profiles, according to the Debye–Scherrer equation (Equation (1)). The strong reflection of [002] was used by measuring the full width at half maximum (FWHM).
(1)$$D=\frac{k\lambda }{\beta cos\theta }\;$$
where *K* is the Scherrer constant with a value of 0.9 [30], λ is the wavelength of light used for diffraction, *β* is the “full width at half maximum (FWHM)” of the [002] peak, and *θ* is the measured angle.

#### 2.2.2. Fourier Transform Infrared Spectrometry

The chemical composition of the powders was analyzed using Fourier Transform Infrared Spectroscopy (FTIR; Bruker Tensor 27 spectrometer; Bruker Corporation, Billerica, MA, USA). FTIR spectra were recorded in Attenuated Total Reflectance (ATR) mode. Spectra were obtained at a resolution of 4 cm^−1^, over a range of wavenumbers from 400 cm^−1^ to 4000 cm^−1^, with an average of 50 scans. Before every measurement, a background spectrum was taken and deducted from the sample spectrum.

#### 2.2.3. Specific Surface Area and Particle Size

The specific surface area (SSA) of the powders was determined using the Brunauer–Emmett–Teller (BET) method (ISO 9277:2010; QUADRASORB SI and Quadra Win, Quantachrome Instruments, Boynton Beach, FL, USA). Before BET analysis, samples were degassed for 24 h at 25 °C (Autosorb Degasser Model AD-9; USA) to remove all moisture and vapor. The SSA of the samples was analyzed using a nitrogen adsorption–desorption isotherm.

Particle size d_BET_ was calculated according to Equation (2) as stated in ISO standard No. 13779-3 “Implants for surgery Hydroxyapatite Part 3: Chemical analysis and characterization of crystallinity and phase purity”, assuming particles to be spherical and nonporous.
(2)$$d_{BET}=6/\left(\rho \times SSA\right)$$
where *ρ* is the density of HAp and GaHAp, determined with a helium pycnometer (Micro UltraPyc 1200e; Quantachrome Instruments, Boynton Beach, FL, USA) as described in Section 2.2.4.

#### 2.2.4. Helium Pycnometry

The true density of the powders was determined using a helium pycnometer. The instrument (cell volume) was calibrated with stainless-steel calibration spheres of known volume. After calibration, samples with known weight were filled into the sample cell and purged with helium gas in pulse mode (50 pulses). Detailed measurement parameters for helium pycnometry are described elsewhere [35].

#### 2.2.5. Transmission Electron Microscope 

The morphology of the powders was observed using a transmission electron microscope (TEM; FEI Tecnai G2 F20; Hillsboro, OR, USA) operated at 200 kV. Detailed sample preparation for TEM analysis is described elsewhere [36].

### 2.3. In Vitro Release of Gallium Ions

The investigation of Ga^3+^ ion release from GaHAp paste was performed in Dulbecco’s Modified Eagle Medium (DMEM) with 1 g/L glucose (without NaHCO_3_; Gibco, Thermo fiscer science, Waltham, MA, USA), with the addition of NaHCO_3_ (99.7%; Sigma-Aldrich, Burlington, MA, USA) and NaN_3_ (99.5% (as preservative); Sigma-Aldrich, Burlington, MA, USA). Afterwards, the medium was filter-sterilized through a 0.22 μm filter.

Prior to the ion release tests, GaHAp paste was steam-sterilized at 121 °C for 20 min. The sterile paste samples (50 mg of dry mass) were added to 50 mL of DMEM, vortexed and incubated at 37 °C in a table-top environmental shaker–incubator at 70 rpm (ES-20; Biosan, Riga, Latvia). During the first 72 h, the medium was collected by centrifuging the samples at 1610 g for 3 min and was then replaced with 50 mL of fresh DMEM every 24 h. Thereafter, the medium was refreshed every 72 h. The Ga concentration in the eluate was measured using ICP-MS (Agilent 7700X; Santa Clara, CA, USA). Three parallel measurements were performed for each GaHAp paste composition.

### 2.4. Antibacterial Tests

The antibacterial properties of GaHAp and Ga(NO_3_)_3_·4.2H_2_O were determined against five bacterial species: Gram-negative *P. aeruginosa* (strain Paer09) and *E. coli* (strain American Type Culture Collection (ATCC) 25922); Gram-positive *S. aureus* (strain JAR 06013), *S. epidermidis* (strain ATCC 35984), and *S. pyogenes* (strain ATCC 19615). Different bacterial species were recovered from frozen stocks (−80 °C in 20% (*v/v*) glycerol) and cultured in tryptic soy broth (TSB; Oxoid, Basel, Switzerland) overnight in ambient air at 37 °C and agitation at 100 rpm. The overnight culture was then diluted with TSB to an optical density (OD) of 0.1 at 600 nm (10^6^–10^7^ colony-forming units (CFU)/mL).

The GaHAp powder used for the antibacterial experiments was prepared from sterilized paste that was dried for 24 h at 105 °C and ground using a pestle. Afterwards, the powder was packed and sterilized with hot air in a drying oven for 2 h at 134 °C.

#### 2.4.1. Minimal Inhibitory Concentration (MIC) of Ga(NO_3_)_3_·4.2H_2_O

Ga(NO_3_)_3_·4.2H_2_O was dissolved in milliQ water and diluted from 75 μg/mL to 450 μg/mL with TSB. A total volume of 150 μL of the solutions was mixed with 150 μL of TSB in a 96-well plate. A volume of 5 μL of bacterial culture with OD_600_ = 0.1 was added to each well and incubated for 24 h at 37 °C at 100 rpm. Bacterial growth (OD_600_) was measured for 18 h at 37 °C in a plate reader (MultiskanGo; Thermo Scientific, Waltham, MA, USA) or it was quantified via serial dilution and total viable count on tryptic soy agar (TSA) plates.

#### 2.4.2. Antibacterial Properties of GaHAp

GaHAp powders were suspended in TSB at concentrations of 1, 2, and 4 mg/mL. A total of 300 μL of the suspension was transferred to a 96-well plate, and 5 μL (OD_600_ = 0.1) of bacteria was added to each well. The absorbance of the plate at 600 nm was measured as described in Section 2.4.1.

### 2.5. Cytocompatibility Test

#### 2.5.1. Cytotoxicity of Ga(NO_3_)_3_·4.2H_2_O

Cytotoxicity was tested on telomerase-immortalized human foreskin fibroblasts (hTERT-BJ1), which were purchased from Clontech (Clontech Laboratories, Mountain View, CA, USA). hTERT-BJ1 were routinely cultured as previously described [37]. Briefly, hTERT-BJ1 were cultivated in DMEM with 1 g/L glucose (without NaHCO_3_; Gibco, Thermo fiscer science, Waltham, MA, USA), supplemented with NaHCO_3_ (99.7%; Sigma-Aldrich, Burlington, MA, USA) and 10 % fetal bovine serum (Biochrome, Sigma-Aldrich, Burlington, MA, USA), with the addition of 100 μg/mL streptomycin (Gibco) and 100 U/mL penicillin (Gibco), at 37 °C in a humidified 5% CO_2_ atmosphere. In total, 10^4^ cells per well were seeded on a 96-well plate. The following day, Ga(NO_3_)_3_·4.2H_2_O solutions at different concentrations (75–450 μg/mL) were applied to the cells and incubated for one and three days. To determine cell viability, CellTiter-Blue (Promega, Promega Corporation, Madison, WI, USA) was performed following the manufacturer’s instructions. Cell viability (calculated in %) was determined as the fluorescence ratio between cells grown in the presence and absence of Ga(NO_3_)_3_·4.2H_2_O solutions. The average values and standard deviations were calculated from three parallel samples in three independent experiments. Dimethyl sulfoxide (DMSO; Sigma-Aldrich, Burlington, MA, USA) was used as a negative control.

#### 2.5.2. Cytotoxicity of GaHAp

hTERT-BJ1 cells were cultured as described above, and the cytotoxicity of GaHAp was assessed using direct and indirect methods. For the direct test, 10^4^ cells per well were seeded on a 96-well plate, and the following day, GaHAp powder suspensions (prepared as described in Section 2.4.2, at concentrations of 1, 2, and 4 mg/mL in DMEM) were applied to the cells and incubated for one, three, and seven days.

In the indirect test, 1.5 × 10^4^ cells per well were seeded on a six-well plate, and the following day, a cell strainer (Corning^®^, Amsterdam, The Netherlands), with a pore size of 100 µm containing GaHAp paste (250 ± 50 mg), was placed in each well and incubated for one, three, and seven days. In order to determine cell viability, CellTiter-Blue was performed in both tests following the manufacturer’s instructions. Cell viability (%) was determined as the fluorescence ratio between cells grown in the presence and absence of GaHAp. The average values and standard deviations were calculated from three parallel samples. As a negative control, DMSO was used.

### 2.6. Statistical Analysis

The results are presented as mean values ± standard deviations (SDs) of three experiments. Statistical analysis was performed on microstructure parameters (specific surface area, density, and particle size) and on cytocompatibility test data using one-way ANOVA with Tukey’s multiple comparison test, and *p* < 0.05 was used as a limit to indicate statistical significance (ns > 0.05; * *p* < 0.05; ** *p* < 0.01; *** *p* < 0.005; **** *p* < 0.001).

## 3. Results

### 3.1. Physicochemical Characteristics

The main characteristics of the synthesized powders are summarized in Table 1. The GaHAp powders had a higher specific surface area (SSA) than the HAp powders (*p* < 0.05), revealing that the addition of Ga led to the reduction in the particle size. After steam sterilization, the SSA decreased. Nevertheless, the SSA of the sterilized GaHAp powders was higher than that of the HAp powders (*p* < 0.05).

The phase composition of the synthesized powders was analyzed using XRD, and the corresponding XRD patterns are shown in Figure 1. Regardless of the chemical composition, all the XRD patterns have characteristic apatite peaks that correspond to the values reported in the literature [38,39]. The XRD patterns did not reveal the presence of additional phases or peak shifts across the different materials. The different amounts of Ga added to the synthesis (2, 4, 6.3, or 8 wt%) did, however, produce a broadening of the XRD peaks (Figure 1A). After steam sterilization, the characteristic peaks become sharper, which indicates an increase in the crystallinity of the samples (Figure 1B).

The crystallite sizes of the diffraction plane [002] for HAp and GaHAp powders were calculated using the Debye–Scherrer equation, and the obtained values are summarized in Table 2. According to these calculations, nano-sized crystallites were obtained, regardless of their chemical composition. With the addition of Ga, the crystallite size of GaHAp decreased compared with the HAp powders. However, before sterilization, differences in crystallite size were not observed among the GaHAp powders.

The FTIR spectra (Figure 2) of HAp and GaHAp show similar appearances. All spectra have HAp characteristic absorbance bands corresponding to ν_3_ PO_4_^3−^ group vibrations at 1030 and 1097 cm^−1^. The absorbance band from the vibration of group ν_1_ PO_4_^3−^ is observed at 960 cm^−1^. Moreover, the absorbance bands at 604 and 560 cm^−1^ can be attributed to ν_4_ PO_4_^3−^ group vibrations. The band at 635 cm^−1^ corresponds to the vibration of the OH^−^ group. The band assignments are in accordance with literature data [40,41]. The characteristic absorbance bands become broader with the increase in Ga content in the synthesized powders.

TEM micrographs were used to analyze the morphology of HAp and GaHAp particles (Figure 3). The TEM results showed a rod-like shape for GaHAp particles and a size less than 50 nm, which corresponds to the value calculated using BET data. The size of the nanoparticles decreased with the increase in Ga concentration.

### 3.2. In Vitro Ga^*3*+^ Release

The release profiles of Ga^3+^ from the GaHAp paste samples are shown in Figure 4. For 2 GaHAp and 4 GaHAp, ion release was measured until days 21 and 27, respectively, as the amount of Ga detected at subsequent time points was below the ICP-MS detection limit (<0.2 mg/kg). In the case of samples with higher Ga concentrations, i.e., 6.3 GaHAp and 8 GaHAp, a gradual release was observed up to 30 days. No high initial or burst release of Ga^3+^ was observed for any GaHAp. The cumulative Ga^3+^ release rate was higher from samples with lower Ga content, namely, 2 GaHAp (44.3 ± 0.9%) and 4 GaHAp (43.1 ± 3.6%), while 6.3 GaHAp and 8 GaHAp Ga release rates were 48.6 ± 1.1% and 49.8 ± 4.6%, respectively, within 30 days.

### 3.3. Antibacterial Activity

#### 3.3.1. Minimal Inhibitory Concentration of Ga(NO_3_)_3_·4.2H_2_O

*P. aeruginosa* and *S. aureus* growth in the presence of Ga(NO_3_)_3_·4.2H_2_O solution was measured by means of absorbance (OD_600_), and the results are shown in Figure 5. After 18 h of incubation, a total inhibition of *P. aeruginosa* growth was observed (Figure 5a). In contrast, *S. aureus* (Figure 5b) and *E. coli*, *S. epidermidis*, and *S. pyogenes* (Appendix A) showed growth reduction in the presence of Ga(NO_3_)_3_·4.2H_2_O, but not complete inhibition.

The minimal inhibitory concentration (MIC) and CFU/mL of bacterial species are shown in Appendix A. The MIC of Ga(NO_3_)_3_·4.2H_2_O against *P. aeruginosa* was 75 μg/mL; against *S. aureus*—150 μg/mL; against *E. coli*—200 μg/mL; against *S. epidermidis*—250 μg/mL; and against *S. pyogenes*—75 μg/mL.

#### 3.3.2. Antibacterial Properties of GaHAp

The growth curves of *P. aeruginosa* and *S. aureus* in the presence of 1, 2, and 4 mg/mL GaHAp powder suspensions are shown in Figure 6 and Figure 7. The growth curves of the other bacterial species (*S. epidermidis*, *S. pyogenes*, and *E. coli*) are shown in Appendix A. 

In the case of 1 mg/mL GaHAp, only the highest Ga-containing powder (8 GaHAp) showed inhibitory effects on *P. aeruginosa* growth (Figure 6a). With 2 mg/mL (Figure 6b) and 4 mg/mL (Figure 6c) GaHAp, a stronger inhibitory effect was observed, and when using 4 mg/mL (Figure 6c), total inhibition of *P. aeruginosa* growth with 4 GaHAp, 6.3 GaHAp, and 8 GaHAp was observed. Growth inhibition was detected with the same concentration of 4 GaHAp and 6 GaHAp against *S. aureus* (Figure 7c), *S. epidermidis* (Appendix A), *S. pyogenes* (Appendix A), and *E. coli* (Appendix A).

### 3.4. Cytocompatibility Test

#### 3.4.1. Cytotoxicity of Ga(NO_3_) _3_·4.2H_2_O

The cell viability of hTERT-BJ1 in the presence of Ga(NO_3_)_3_·4.2H_2_O solution is shown in Figure 8. A rapid decrease in cell viability was observed after the first day, starting from 150 μg/mL Ga(NO_3_)_3_·4.2H_2_O (*p* < 0.001). Concentrations of Ga(NO_3_)_3_·4.2H_2_O above the MIC (75 μg/mL) exhibited an increased toxicity already on the first day. After three days, only the cells exposed to 75 µg/mL Ga(NO_3_)_3_·4.2H_2_O showed maintained metabolic activity, around 80%.

#### 3.4.2. Cytotoxicity of GaHAp

The influence of GaHAp on human fibroblast (hTERT-BJ1) viability was tested by applying GaHAp powder directly on the cells. After three days of exposure to different GaHAp concentrations (1, 2, and 4 mg/mL), cells were still metabolically active (above 80% viability) (Figure 9b). However, on day seven (Figure 9c), a rapid decrease in cell viability was observed with 2 mg/mL 8 GaHAp and with 4 mg/mL 4 GaHAp, 6.3 GaHAp, and 8 GaHAp paste samples (less than 40 %).

Human fibroblast (hTERT-BJ1) viability using the different GaHAp paste samples is shown in Figure 10. The indirect test results indicate that the released Ga^3+^ from GaHAp paste samples did not have cytotoxic effect at any time point, and even after seven days, cell viability was around 90%. Additionally, the metabolic activity of the cells exposed to 6.3 GaHAp and 8 GaHAp paste samples on day three was significantly increased.

## 4. Discussion

In the present work, we successfully synthesized Ga-doped-HAp using the wet chemical precipitation method.

The morphological and structural nature of the obtained GaHAp coincides with that in previously described studies [29,33,42,43]. The synthesized products had low crystallinity, as suggested by the low-intensity, broad XRD peaks [38,39]. The adsorption of smaller ions, such as Ga^3+^ compared with Ca^2+^, on the HAp crystal surface results in the inhibition of crystallization and crystal growth [29,42,44,45,46]. Additionally, steam sterilization had a significant effect on the morphology of the final product. Re-crystallization of the product was observed in the XRD patterns, due to the fact that a hydrothermal reaction occurs under high-temperature and -pressure conditions. This process leads to the nucleation of the crystals and their growth as described in [44,45]. Due to the low crystallinity of the samples obtained in our study, it is challenging to detect the Ga^3+^ substitution of Ca^2+^ in the structure. Possibly, Ga^3+^ is adsorbed/chemisorbed on the particle surface or taken up in the interstitial positions [47]. However, there is no universal technique to detect ion substitution in the interstitial position.

The potential of HAp materials to act as delivery systems of antibacterial Ga^3+^ ions was investigated by assessing ion release in the cell culture medium. The release of antibacterial ions from the implant material during the initial periods after surgery is important in preventing the development of infections [48]. As bone-associated infections oftenoccur within four months of surgical interventionsprolonged delivery of antibacterial ions may be necessary [28]. Thus, it is important to evaluate the release of Ga^3+^ ions over long periods. The Ga^3+^ release profile depends on the Ga content in products, the degree of Ca deficiency of products, the released media, and the conditions [49]. The increase in Ga content in the HAp samples is at the expense of reducing the Ca/P molar ratio. Thus, non-stoichiometric or Ca-deficient HAp (CDHAp) is obtained, which is more soluble than HAp [50]. If Ga does not enter the Ca site in HAp crystallites, it accumulates in the hydrated layer on the surface of crystallites. Additionally, during the first 72 h, the cumulatively released amount of Ga^3+^ increases more rapidly, i.e., the frequent refreshing of the medium results in a faster release of the ions. Ideally, the flow of the release medium should be aligned with the flow rate of physiological fluids at the site of implantation. The results of the ion release tests have shown that by increasing the Ga concentration above 3.3 ± 0.4 wt%, it is possible to obtain HAp for long-term delivery of Ga^3+^. Furthermore, approximately 50% of ions were not released within the timeframe of our study, suggesting further long-term delivery up to four months.

The biological properties of Ga(NO_3_)_3_·4.2H_2_O, used as the gallium source, were compared to GaHAp paste or GaHAp powder. Ga(NO_3_)_3_·4.2H_2_O and GaHAp showed similar antibacterial activity against different Gram-positive and Gram-negative bacterial species. This might be connected to the bacterial cell wall structure (Appendix A) [32,51]. Gram-positive bacteria have a thicker cell wall, which makes the bacterial cell impenetrable for Ga compared with Gram-negative bacteria. On the other hand, Gram-negative bacteria, with their thinner cell wall, have a membrane that can lead to Ga penetration. Additionally, Gram-negative bacteria have Fe-dependent metabolism, and Ga^3+^ can replace Fe^3+^ on active enzymatic sites and disrupt protein metabolism, leading to bacterial death [52]. Interestingly, the HAp nanoparticles without Ga also showed delayed bacterial growth. This could be explained by the fact that nanoparticles themselves can have a negative effect on bacterial growth [48,50]. However, GaHAp did not show total inhibition of *P. aeruginosa* growth, as observed in the case of Ga(NO_3_)_3_·4.2H_2_O. This effect is related to the delayed Ga^3+^ release from HAp, leading to lower Ga^3+^ concentration in media during the first 24 h. Ga(NO_3_)_3_·4.2H_2_O completely inhibited *P. aeruginosa* growth, but this was not observed in the case of *E. coli*, even though both bacteria are Gram negative. The iron uptake pathway via siderophore enterobactin (ENT) in *P. aeruginosa* and *E. coli* is different [53]. In contrast with other studies, in our results, we obtained bacterial growth inhibition at higher concentrations of GaHAp powder. For example, Kurtjak et al. obtained inhibition of *P. aeruginosa* growth with 0.9 g/mL GaHAp containing 3 wt% of Ga (synthesized using the co-precipitation method) [29]. In addition, Ballardini et al. showed an antibacterial effect of Ga-doped HAp against *P. aeruginosa* and *S. aureus*. However, *E. coli* and *C. albicans* showed higher resistance to Ga-doped HAp after 24 h [54]. Even though the final product was Ga-doped-HAp, the synthesis methods used differed in the previous examples. These important findings indicate that the synthesis method influences key properties of the final material.

It is also important to test the biocompatibility of the produced material. In this study, we observed different effects of Ga(NO_3_)_3_·4.2H_2_O and GaHAp paste or powder on cell metabolic activity. Ga(NO_3_)_3_·4.2H_2_O has a strong acidic nature, which hydrolyzes in a wide pH range of aqueous media. This process leads to the formation of hydroxylate species, predominantly [Ga(OH)_4_]^−^ and hydronium ion (H_3_O^+^) [55,56,57], leading to the acidification of cell culture media. Indeed, we observed a change in medium color between 75 and 450 μg/mL Ga(NO_3_)_3_·4.2H_2_O, indicating a decrease in pH, which results in cell death. The results from the direct and indirect tests of GaHAp on human fibroblast show the importance of evaluating the interactions between the new developed biomaterial and cells. When the materials (GaHAp paste or powder) were not in direct contact with the cells, we observed a higher metabolic activity in fibroblasts after 7 days. Presumably, the ions released from GaHAp paste (Ga^3+^ but also Ca^2+^) can stimulate cell growth. In the case of pure HAp, cell viability was approximately 120%. It has already been reported that Ca^2+^ ions promote bone formation and maturation [2]. From the literature, Pajor et al. observed the toxic effect of GaHAp prepared with the dry method on the BALB/c 3T3 clone A31 mammalian cell line compared with the same material prepared with the wet method. In the study, this coherence was explained with the solubility of the materials obtained with different synthesis methods [42]. That was another confirmation that the material form and the method used can have an influence on material–cell interactions.

## 5. Conclusions

Gallium-doped hydroxyapatite (GaHAp) was successfully obtained, and it showed promising biological properties. The optimal Ga^3+^ doping rate of HAp ranges from 2 to 5.5 ± 0.1 wt%, as it was shown from the analyses of the morphological properties and biological activity. The addition of Ga to the synthesis media promoted the formation of HAp with smaller particle sizes. GaHAp provided long-term gallium ion release with Ga concentrations above 3.3 ± 0.4 wt%. Additionally, GaHAp had a bacteriostatic effect on multiple bacterial species, both Gram positive and Gram negative, without substantial toxicity towards human fibroblasts. The GaHAp samples showed superior inhibition of *P. aeruginosa* compared with the other bacterial species, representing a material advantage for the early-stage treatment of bone defect, as it prevents further bacterial growth and could prevent the development of chronic and acute infection.

## Figures and Tables

**Figure 1 jfb-14-00051-f001:**
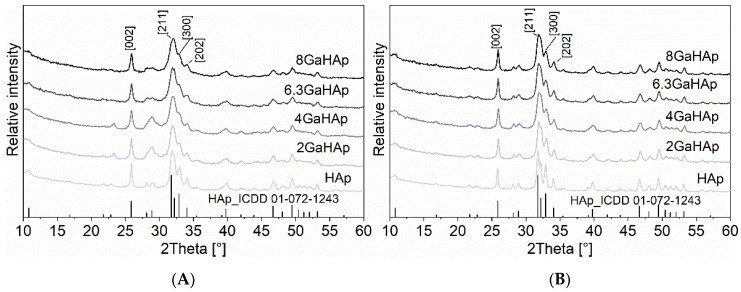
X-ray diffraction patterns of the synthesized powders with different amounts of gallium (**A**) before and (**B**) after steam sterilization at 121 °C for 20 min.

**Figure 2 jfb-14-00051-f002:**
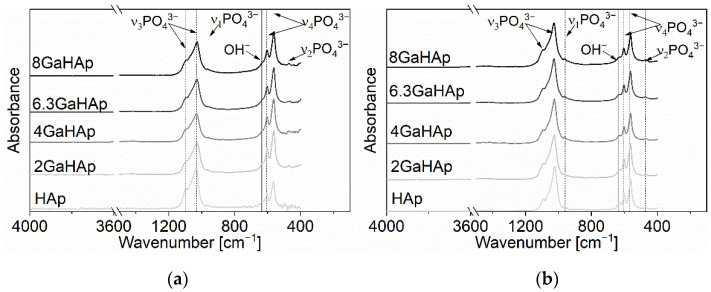
FTIR spectra of the synthesized powders with different amounts of gallium (**a**) before and (**b**) after steam sterilization at 121 °C for 20 min.

**Figure 3 jfb-14-00051-f003:**
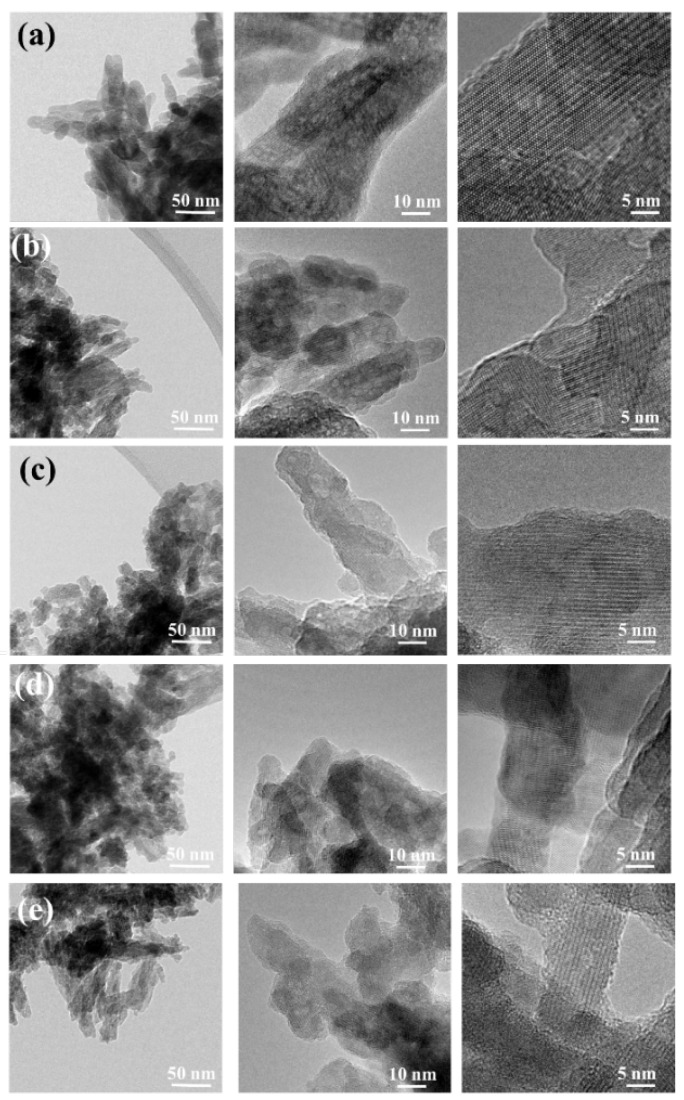
TEM images of synthesized GaHAp powders with different amounts of gallium: (**a**) HAp (**b**) 2 GaHAp; (**c**) 4 GaHAp; (**d**) 6.3 GaHAp; (**e**) 8 GaHAp.

**Figure 4 jfb-14-00051-f004:**
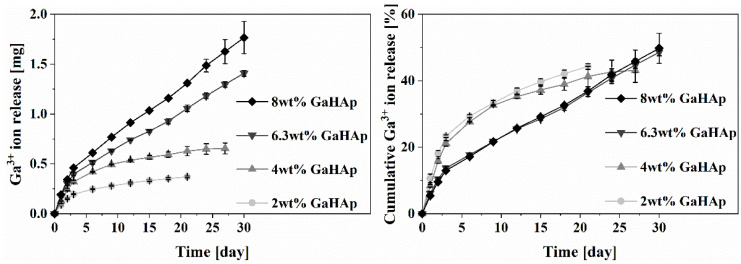
Ga^3+^ ion release from GaHAp paste with different amounts of gallium as a function of a time. Release overtime was performed in DMEM media at 37 °C (*n* = 3 ± SD).

**Figure 5 jfb-14-00051-f005:**
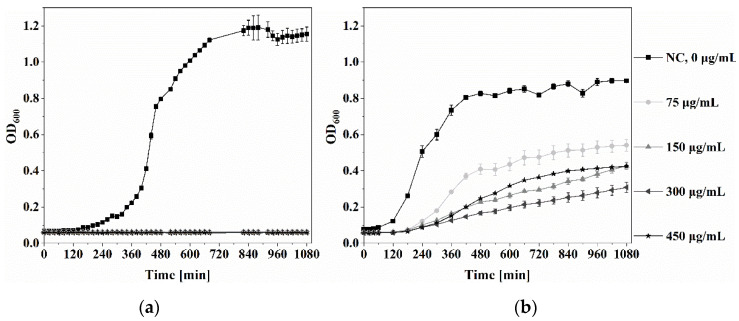
Bacterial growth of *P. aeruginosa* and *S. aureus* in the presence of Ga(NO_3_)_3_·4.2H_2_O. (**a**) *P. aeruginosa* and (**b**) *S. aureus* were grown in TSB with different concentrations of Ga(NO_3_)_3_·4.2H_2_O (from 0 µg/mL to 450 µg/mL Ga(NO_3_)_3_·4.2H_2_O ). The optical density (OD) at 600 nm was measured over 18 h using a plate reader.

**Figure 6 jfb-14-00051-f006:**
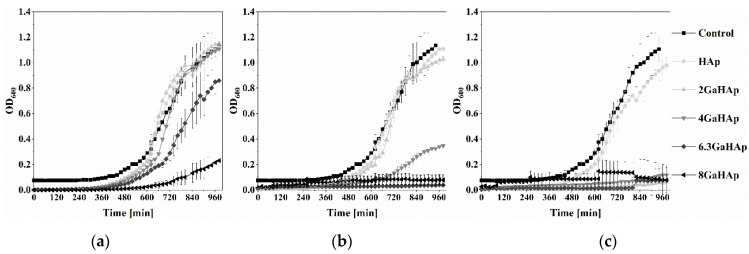
*P. aeruginosa* growth in the presence of GaHAp powder suspensions in TSB at (**a**) 1 mg/mL (**b**) 2 mg/mL, and (**c**) 4 mg/mL. The represented OD_600_ was obtained by subtracting the initial OD_600_oh_ (starting time = 0 h) from the measured OD_600_x h_ at specific time points.

**Figure 7 jfb-14-00051-f007:**
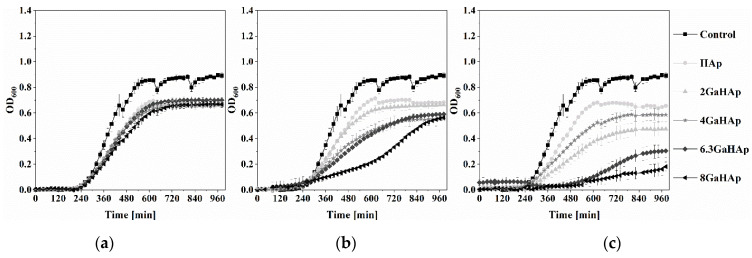
*S. aureus* growth inhibition induced by GaHAp powder suspensions in TSB at (**a**) 1 mg/mL (**b**) 2 mg/mL, and (**c**) 4 mg/mL concentrations. The represented OD_600_ was obtained by subtracting the initial OD_600_oh_ (starting time = 0 h) from the measured OD_600_x h_ at specific time points.

**Figure 8 jfb-14-00051-f008:**
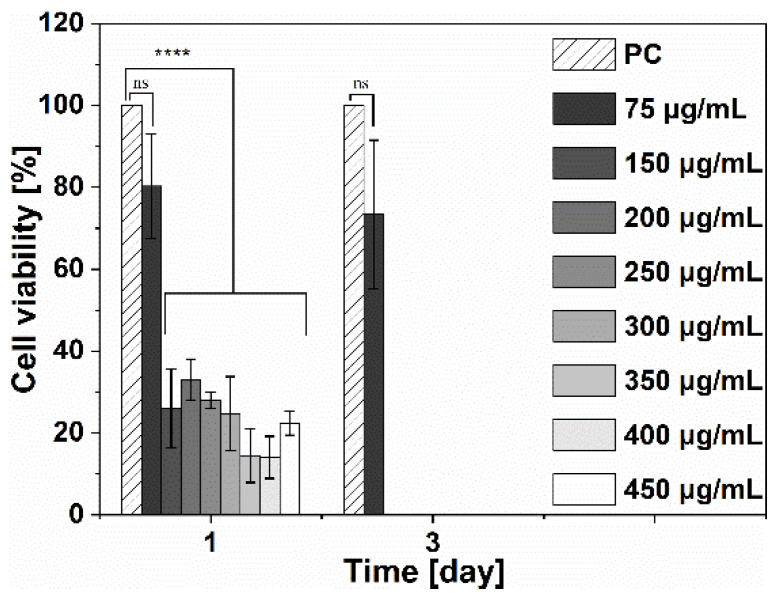
Metabolic activity of human fibroblasts (hTERT-BJ1) exposed to different concentrations of Ga(NO_3_)_3_·4.2H_2_O solutions, where PC—positive control. Negative control results were omitted from the graphs as the values were approximately 0%. (One-way ANOVA single factor with Tukey’s multiple comparison test, *n* = 3; ns > 0.05; **** *p* < 0.001).

**Figure 9 jfb-14-00051-f009:**
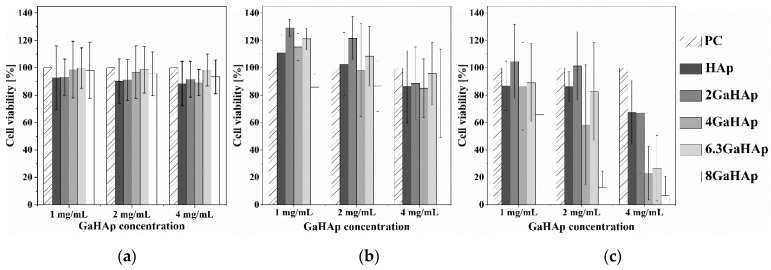
Metabolic activity of human fibroblasts (hTERT-BJ1) exposed to GaHAp powder suspensions at different concentrations (1, 2, and 4 mg/mL) obtained using the direct test: (**a**) day 1; (**b**) day 3, and (**c**) day 7. PC—positive control (ANOVA test, *n* = 3).

**Figure 10 jfb-14-00051-f010:**
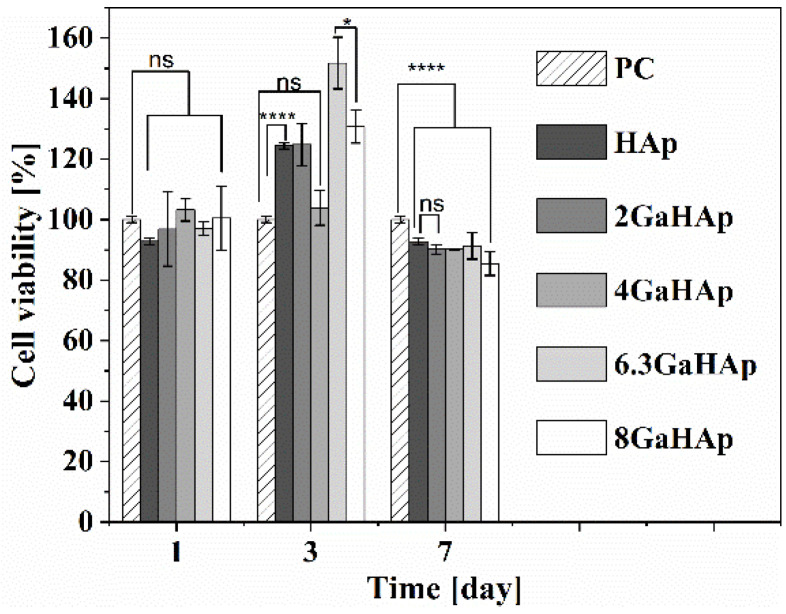
Metabolic activity of human fibroblasts (hTERT-BJ1) exposed to Ga^3+^ released from GaHAp paste obtained using the indirect test. (One-way ANOVA single factor with Tukey’s multiple comparison test, *n* = 3; ns > 0.05; * *p* < 0.005; **** *p* < 0.001).

**Table 1 jfb-14-00051-t001:** Ga contents, particle sizes (d_BET_), densities, and SSAs of the synthesized products before and after sterilization.

Sample	Theoretical Ga Content (wt%)	Gallium Content (wt%)	Before Sterilization			After Sterilization		
			SSA (m^2^/g)	ρ (g/cm^3^)	d_BET_ (nm)	SSA (m^2^/g)	ρ (g/cm^3^)	d_BET_ (nm)
HAp	-	-	71 ± 7	2.77 ± 0.04	31 ± 3	50 ± 2	2.94 ± 0.01	41 ± 2
2 GaHAp	2	1.6 ± 0.1	95 ± 5	2.85 ± 0.06	22 ± 1	78 ± 3	2.86 ± 0.08	27 ± 1
4 GaHAp	4	3.3 ± 0.4	117 ± 4	2.80 ± 0.06	18 ± 1	88 ± 5	2.92 ± 0.09	23 ± 1
6.3 GaHAp	6.3	5.5 ± 0.1	109 ± 2	2.83 ± 0.01	20 ± 1	89 ± 4	2.92 ± 0.05	23 ± 1
8 GaHAp	8	6.9 ± 0.5	102 ± 5	2.79 ± 0.03	21 ± 1	104 ± 3	2.84 ± 0.04	20 ± 1

**Table 2 jfb-14-00051-t002:** Crystallite sizes of the synthesized products calculated using the Debye–Scherrer equation.

Sample	Plane (hkl)	Before Sterilization		After Sterilization	
		FWHM	Crystallite Size (nm)	FWHM	Crystallite Size (nm)
HAp	[002]	0.274	31.1	0.219	38.9
2 GaHAp	[002]	0.347	24.5	0.284	30.0
4 GaHAp	[002]	0.384	22.2	0.279	30.5
6.3 GaHAp	[002]	0.383	22.2	0.309	27.6
8 GaHAp	[002]	0.372	22.9	0.316	27.0

## Data Availability

The data presented in this study are available upon request from the corresponding author. The data are not publicly available due to project agreement.

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
