# Peer review of "Gallium-Doped Hydroxyapatite Shows Antibacterial Activity against *Pseudomonas aeruginosa* without Affecting Cell Metabolic Activity"

_jfb, 2023, doi:10.3390/jfb14020051_

Round 1

Reviewer 1 Report

Generally, the results are well discussed providing consistent explanations. The manuscript is well prepared and brings interesting results. The characterization methods and methodology used to assess the antimicrobial activity are adequate.

I have only some minor corrections:

Line 280: “(43.1±3.6%), while 6.3GaHAp” instead of “(43.1±3.6%). While 6.3GaHAp”.

Line 413: “the” before “a wide pH range of aqueous media” must be deleted.

Line: 414: [Ga(OH)4]-  is not gallium gallate. Please correct!

Author Response

Authors reply to the reviewer’s comments:

Reviewer 1

Generally, the results are well discussed providing consistent explanations. The manuscript is well prepared and brings interesting results. The characterization methods and methodology used to assess the antimicrobial activity are adequate.

I have only some minor corrections:

Line 280: “(43.1±3.6%), while 6.3GaHAp” instead of “(43.1±3.6%). While 6.3GaHAp”.

Line 413: “the” before “a wide pH range of aqueous media” must be deleted.

Line: 414: [Ga(OH)4]-  is not gallium gallate. Please correct!

Authors response: Thank you for the suggestions. The Authors have made the needed corrections on the lines 280, 413 and 414.

Reviewer 2 Report

The reviewed manuscript investigates that gallium doped hydroxyapatite shows antibacterial activity against pseudomonas aeruginosa without affecting cell meta- bolic activity. Authors demontrated that gallium doped hydroxyapatite (GaHAp) was successfully obtained and showed promising biological properties.

1.       The introduction needs a lot of enhancement. I think it needs to be extended to illustrate the novelty and the importance of the work, as well as some previous related studies in published literatures. After all, the idea that the antibacterial activity of GaHAp is not a new one, which had been reported against different bacterial species. This paper is not innovative enough. The highlights of this work are very common, no breakthrough.

2.      The mechanism of some results is insufficient. A more profound discussion is required to improve the readability and clarity of the manuscript.

Author Response

Authors reply to the reviewer’s comments:

Reviewer 2

The reviewed manuscript investigates that gallium doped hydroxyapatite shows antibacterial activity against pseudomonas aeruginosa without affecting cell meta- bolic activity. Authors demontrated that gallium doped hydroxyapatite (GaHAp) was successfully obtained and showed promising biological properties.

  1. The introduction needs a lot of enhancement. I think it needs to be extended to illustrate the novelty and the importance of the work, as well as some previous related studies in published literatures. After all, the idea that the antibacterial activity of GaHAp is not a new one, which had been reported against different bacterial species. This paper is not innovative enough. The highlights of this work are very common, no breakthrough.

Authors’ response: The Authors would like to thank for the insight. There have indeed been studies on similar materials before (Kurtjak, M. et al., RSC Adv. 2016; Pajor, K. el al., Int. J. Mol. Sci. 2020; Melnikov, P. Mater. Lett. 2019). Having this in mind, we have conducted a systematic study and showed that, to an extent, some of the results have contradicted the ones published thus far. For example, the published results showed the materials bactericidal effect against P. aeruginosa, however, we have proved only bacteriostatic activity against P. aeruginosa, where bacterial growth was stopped. Providing the additional findings of the importance of the appropriately chosen synthesis method on the antibacterial potential is incremental for the scientific community.

  1. The mechanism of some results is insufficient. A more profound discussion is required to improve the readability and clarity of the manuscript.

Authors’ response: The authors would like to sincerely thank the Reviewer for the comments. The Authors have revised the manuscript for the improvement of readability and accentuated some of the findings of the present manuscript. However, the Authors believe that throughout the discussion section, the results attained from the phase and composition characterization, followed by the in vitro assays have been complementary and they have been presented in an orderly manner. Furthermore, the possibility to compare the different sets of data from the multi-technique approach is presented with the aim that the reader from any background can easily conclude what is the difference once Ga was added in the HAp materials and which benefits have resulted from it.

Reviewer 3 Report

The article is original and very interesting. I recommend few minor revisions.

1. As supplementary proof of your results, you may add some photos of cell cultures surviving at biomaterials exposure

2. A thorough check by a native English speaker ( for grammar and spelling mistakes) would be very ussefull

3. Follow strictly the Instructions for authors, both in the text and in References section (few of it don t have doi code)

Author Response

Authors reply to the reviewer’s comments:

Reviewer 3

The article is original and very interesting. I recommend few minor revisions.

  1. As supplementary proof of your results, you may add some photos of cell cultures surviving at biomaterials exposure

Authors’ response: Thank you for your suggestion. The Authors have added photos of the cells exposed to the biomaterial in the supplementary data – Figure S6.

  1. A thorough check by a native English speaker ( for grammar and spelling mistakes) would be very ussefull

 Authors’ response: The entire manuscript has been once again revised and the changes can be seen in track changes.

  1. Follow strictly the Instructions for authors, both in the text and in References section (few of it don t have doi code)

Authors’ response: The entire manuscript has been once again revised according to the instructions for the authors.

Round 2

Reviewer 2 Report

The revised version has been well improved. 

 Accept in present form.